# Face mask ownership/utilisation and COVID-19 vaccine hesitancy amongst patients recovering from COVID-19 in Cameroon: A cross-sectional study

**Frederick Nchang Cho**[1,2☯]*, **Yayah Emerencia Ngah**[3☯], **Andrew N. Tassang**[4,5,6], **Celestina Neh Fru**[6,7], **Peter Canisius Kuku Elad**[1,8☯], **Patrick Kofon Jokwi**[1☯], **Valmie Ngassam Folefac**[9,10], **Ismaila Esa**[1], **Paulette Ngum Fru**[11,12]

1 Cameroon Baptist Convention Health Services – HIV free/Strengthening Public Health Laboratory Systems, Buea, Cameroon, 2 Infectious Disease Laboratory, Faculty of Health Sciences, University of Buea, Buea, Cameroon, 3 District Health Services Bamenda, North West Regional Delegation of Health, Ministry of Health, Buea, Cameroon, 4 Department of Obstetrics and Gynaecology, Faculty of Health Sciences, University of Buea, Buea, Cameroon, 5 Buea Regional Hospital Annex, Buea, Cameroon, 6 Atlantic Medical Foundation, Mutengene, Cameroon, 7 Department of Sociology and Anthropology, Faculty of Social and Management Sciences, University of Buea, Buea, Cameroon, 8 Department of Microbiology and Parasitology, University of Buea, Buea, Cameroon, 9 Department of Biochemistry and Molecular Biology, Faculty of Science, University of Buea, Buea, Cameroon, 10 Sintieh Research Academy, Yaoundé, Cameroon, 11 Department of Public Health and Hygiene, Faculty of Health Sciences, University of Buea, Buea, Cameroon, 12 District Health Services Tiko, South West Regional Delegation of Health, Ministry of Health, Buea, Cameroon

☯ These authors contributed equally to this work.
* nchang.cho@gmail.com

## Abstract

### Introduction

This study aimed to establish pre-/post Coronavirus Disease 2019 (COVID-19) diagnosis/treatment symptoms, ownership/utilisation of face masks (FMs), as well as vaccine hesitancy (VH) amongst patients recovering from COVID-19.

### Methods

A cross-sectional survey was conducted from April - October 2021. Data was collected with structured self-administered questionnaires. Multinomial regression was used to determine associations between ownership/utilisation of FMs with respondents' characteristics.

### Results

Unproductive cough and fatigue were prevalent before and after treatment. Pre-/Post COVID-19 symptoms severity ranged from mild to moderate. There was a COVID-19 VH rate of 492 (74%). The prevalence of FM ownership and utilisation were, respectively, 613 (92.2%) and 271 (40.8%). One main factor was associated with FM ownership; respondent's sex ($p$; 5.5x10$^{-2}$, OR; 0.5, 95%C.I; 0.3 – 1.0). The main reasons for irregular utilisation were; inability to be consistent, only used outdoors, and boredom.

**Data Availability Statement:** The data used to support the findings of this study are included within the supplementary information file(s).

**Funding:** The authors received no specific funding for this work.

**Competing interests:** The authors have declared no competing interests exist.

**Abbreviations:** x̄, Mean; COVID-19, Coronavirus disease 2019; FM(s), Face mask(s); SD, Standard Deviation; VH, Vaccine hesitancy; WHO, World Health Organisation; $\chi^2$, Chi-square.

## Conclusion

The treatment of COVID-19 does not mean immediate recovery as mild to moderate grade severity still persists. Face mask availability and ownership does not mean appreciable utilisation. This study advocates for an intensification of COVID-19 preventive practices, as well as elaborate education on the importance of vaccination.

## Introduction

A novel enveloped ribonucleic acid (RNA) beta (β) corona virus, Coronavirus Disease 2019 (COVID-19) has spread widely from Wuhan, causing tens of thousands of deaths, especially in patients with severe COVID-19 as from December 2019 [1–5]. The coronavirus diseases vary from mild, self-limiting forms to more severe manifestations depending on the type of viruses involved [6, 7]. As the pandemic progressed, about 461,175,583 cases, 6,071,057 deaths, and 394,441,157 recovered cases have been reported globally [8]. The impacts of the pandemic have been felt unequally around the world, with Europe and America being highly affected, as shown by overwhelmed health systems and high death tolls [9].

The Coronavirus Disease 2019 (COVID-19) is spreading across Africa, and available data indicates that it is on the rise in Cameroon, Uganda, and other African countries [10, 11]. A third wave of the pandemic, characterised by the delta (δ) strain of the virus, has also reached parts of Africa. The first case of COVID-19 in Cameroon was identified on the 6th of March 2020 [12] and as of 6th July 2020, 320 deaths were recorded and almost 15,000 cases were confirmed [13–15]. As of the 29th of December 2021, 3,756 health personnel tested positive for COVID-19 nationwide [16], of whom two of the 45 from the South West Region of Cameroon died [17], and as the situation progressed 121,650 cases, 1,935 dead, and 117,263 recovered nationwide [8, 18]. As of 28th July 2021, 285,522 people had received the first dose of the vaccine and 53,365 the second, representing 38% consumption of the received vaccines [17].

The reference and definitive diagnosis of Severe Acute Respiratory Syndrome Coronavirus-2 (SARS-CoV-2) infection is the reverse transcription polymerase chain reaction assay (rt-PCR). Chest X-ray (CXR), although not generally considered sensitive for the detection of pulmonary abnormalities in the early stage of the disease, can be a useful diagnostic tool for monitoring the rapid progression of lung involvement in COVID-19, especially in patients admitted to intensive care units (ICUs) [19].

Many approaches have been used for the treatment of high mortality risk patients as well as other patients [20, 21]. The recovery of patients presenting with varying symptoms depends on treatment regimens administered [20], as well as the severity of the infection. The clinical spectrum of COVID-19 varies from asymptomatic presentation (Stage I) to moderate to severe states (Stages IIa and IIb) characterized by respiratory failure necessitating, mechanical ventilation and ICU support and those manifesting with critical clinical condition (Stage III), with mild to moderate disease occurs in approximately in 81% of cases [22].

Individuals of all ages are at risk for infection and severe disease. However, the probability of serious COVID-19 disease is higher in the old, those living in nursing homes or long-term care facilities, and those with chronic medical conditions [21].

In Cameroon, SARS-CoV-2 screening, hydroxychloroqine/azithromycin regimens and COVID-19 vaccines are free in all Health Districts while the ADSAK/ELIXIR COVID costs 20,000Fcfa (30.6€ or 36.3$) and is available in Catholic and Private Health Facilities across the Country. As of 8th July 2021, the following COVID-19 vaccines; Sinopharm BIBP, and Oxford–AstraZeneca were in Cameroon, as well as Janssen (Johnson & Johnson) donated by

the United States Government, and the African Union [16]. A variety of face masks (FMs) are available in Pro-pharmacies in most health facilities, local shops, tailoring shops, and street vendors. The average cost of FMs was 350Fcfa (0.53€ or 0.63$); range 200 – 500Fcfa (0.31 – 0.76€ or 0.36 – 0.91$). The majority of the population using FMs, use the locally made masks/cloth masks that costs 200Fcfa (0.31€ or 0.31$). The monthly income of the population ranges between 16.65 and 72.11 € (or 19.81 and 85.81$) [23]. Conversion rates; 1€ = 655Fcfa, 1$ = 551.13Fcfa.

Cameroon is known as Africa in miniature, with variety of ethnic groups. Apart from the conflict hit areas, Cameroon is an economic hub in Central Africa with an estimated population of about 23,344,000 in 2015 [23]. Due to the pandemic, a couple of studies on COVID-19 have been carried out in Cameroon; vaccine hesitancy/acceptance [24–26], responses to COVID-19 in the educational sector [13], **clinical characteristics and outcomes of patients hospitalised for COVID-19** [27, 28], COVID-19 preventive behaviours [12], as well as knowledge attitudes and practices [29, 30], but none has explored the diagnosis/treatment symptoms, face mask ownership/utilisation, preventive measures, and vaccine hesitancy amongst patients recovering from the COVID-19 infection. Thus, in this study, we aimed to establish pre- and post-COVID-19 diagnosis/treatment symptoms amongst persons who suffered from COVID-19 infection, ownership and utilisation of face masks, as well as vaccine hesitancy.

## Methods

### Study design and setting

A cross-sectional study was conducted from April 29th - July 4th, and August 2nd - October 30th, 2021 amongst persons who had tested positive for COVID-19. Patients who admitted having recovered from COVID-19 as confirmed by a negative control test result, were enrolled into this study. Those who consented to fill the questionnaire, were cautioned not to fill it if they had earlier filled a similar questionnaire, online. Information on pre-/post-recovery clinical symptoms were collected with the use of anonymous online as well as 'hand-filled' questionnaires (S1 Appendix).

Bafoussam, Douala, and Yaoundé are metropolis located respectively in the West, Littoral, and Centre regions of Cameroon, with Douala, and Yaoundé being the largest.

### Study population and target sample size

The study population was persons who by chance had new technology devices and could access social media platforms, and those who were at home/work place/outpatient department (OPD) of selected hospitals. The Google© questionnaire's link was distributed via social media: WhatsApp, LinkedIn, Facebook, Skype, and Instagram. It was also shared from door-to-door, work places and OPD of hospitals in the towns of Bafoussam, Douala, and Yaoundé. Being COVID-19 positive, adult ($\geq$ 20 years of age) and verbal consent to participate in the study constituted the inclusion criteria.

An estimated minimum sample size of 256 per town was calculated with the CDC Epi Info 7 StatCalc, and collected by convenience sampling, considering the following characteristics: an estimated Cameroon population size of 23,344,000 in 2015 [23], an assumed frequency of persons who have recovered from COVID-19 of 50%, accepted error margin of 5%, and design effect of 2.0.

### Sampling method

The study was conducted during the second wave of the COVID-19 pandemic in Cameroon. The convenience sampling technique was implored wherein participants were approached

and informed of the study objectives via social media platforms, at their work places, at their door steps and OPD of hospitals. Investigators continuously posted the questionnaire together with reminders on various platforms, regularly checked incoming responses, as well as regularly checked workplaces, homes, and OPDs to collect 'hand dropped' questionnaires.

## Data collection and analysis

Data was collected with the use of anonymous self-administered online [3, 4] and workplace/door-to-door/OPD questionnaires using Microsoft Office Excel and analysed with CDC Epi Info version 7.2. The questionnaire that consisted of 29 questions, aimed to collect information on: socio-demographic characteristics, ownership/utilisation of FMs, pre-/post-symptoms, and treatment of COVID-19, as well as opinions on COVID-19 vaccination. The survey instrument took approximately 10 minutes to complete. The validity of the questionnaire was confirmed by pre-testing in 10 participants who were excluded from the study. Based on the pre-test study, the format and wording of some questions were corrected and refined. Data from the 10 participants was used to assess internal consistency reliability using Cronbach's alpha ($\alpha$) [31–33]. The results showed adequate internal consistency reliability (with Cronbach's $\alpha = 0.72$) [31, 32].

Age groups, sex, education, and occupation were summarised as counts and percentages. Body Mass Index (BMI), age, household size, number of sleeping rooms, length of hospital stay/quarantine, were expressed as ranges and means. Analysis of variance (ANOVA) and Multinomial Logistic Regression were used to determine associations between hospitalisation/quarantination and ownership/utilisation of FMs with demographic characteristics. In order to control for confounders, the least significant independent covariates were excluded from the Multinomial Logistic Regression analysis. The following models were used:

Ownership of FM = $\beta_0 + \beta_1$Age + $\beta_2$Sex + $\beta_3$Marital Status + $\beta_4$Education + $\beta_5$Occupation + $\beta_6$Family size + $\varepsilon$, and

Household ownership of FM/Correct wearing of FM/ = $\beta_0 + \beta_1$Age + $\beta_2$Sex + $\beta_3$Marital Status + $\beta_4$Education + $\beta_5$Occupation + $\beta_6$Family size + $\beta_7$Residential area + $\varepsilon$. Where $\beta_0$ is a constant, $\beta_1, \beta_2, \beta_3, \ldots$, and $\beta_7$ are coefficients and $\varepsilon$ is the regression error.

The significance level was set at $< 0.05$.

## Definition of concepts/dependent variables

**Severity of COVID-19.** The severity grading of COVID-19 in this study consisted of pre-/post-treatment symptoms and their duration, age ($\geq 60$ years considered riskier), and comorbidities (five; on a score of 2) with a severity grading of $< 20\%$ for asymptomatic, $20 - < 40\%$ for mild, $40 - < 80\%$ for moderate and $\geq 80\%$ for severe COVID-19.

**Ownership of face masks.** The proportion of respondents who had at least one FM, where the numerator comprised the number of respondents surveyed with at least one FM and the denominator, the total number of respondents surveyed. *Entire household ownership of FMs*; the proportion of households wherein all *de facto* members have FMs, where the numerator comprises the number of households wherein all *de facto* members own FMs and the denominator, the total number of respondents surveyed.

**Utilisation of face masks.** *The correct wearing of FMs*; the proportion of respondents who correctly wore FM, where the numerator comprises the number of respondents surveyed correctly wearing FMs as recommended by the WHO [34] and the denominator, the total number of respondents surveyed. *Adequate utilisation of FMs*; the proportion of respondents who wore at least six or more ($\geq 6$) FMs per week, where the numerator comprises the number of respondents surveyed adequately wearing FMs and the denominator, the total number of

respondents surveyed. *Regular utilisation of FMs*; the proportion of respondents who regularly wore their FMs, where the numerator comprises the number of respondents surveyed regularly wearing FMs and the denominator, the total number of respondents surveyed.

**Knowledge related to COVID-19 symptoms.** Knowledge related to symptoms of COVID-19 comprised one open-ended question with the possibility of listing 10 or more options. The options were developed by considering previous studies with a similar research theme [35]. The options were sorted in the form of yes or no; if the answer was yes/no, a score of '1'/'0' was accorded to the participant. Modified Bloom's cut-off points were used to judge knowledge as very poor ($< 20\%$), poor ($20 - < 40\%$), moderate ($40 - < 60\%$), good ($60 - < 80\%$), and very good ($\geq 80\%$) [7].

**Practice regarding COVID-19 prevention.** Prevention practices consisted of the correct and adequate wearing of FMs, and a question on preventive measures towards COVID-19 (one correct point listed by the respondent earned a score of 1 and an omission or wrong point listed earned a score of 0). Modified Bloom's cut-off points were used to judge knowledge as very poor ($< 20\%$), poor ($20 - < 40\%$), moderate ($40 - < 60\%$), good ($60 - < 80\%$), and very good ($\geq 80\%$) [7].

**COVID-19 Vaccine hesitancy.** The proportion of respondents who will not voluntarily take COVID-19 vaccine, where the numerator comprises the number of respondents who will not take the vaccine [36] and the denominator, the total number of respondents surveyed.

## Ethical consideration

This study was conducted in accordance with the Helsinki Declaration [37] as well as the principles of Personal Information Protection and Electronic Documents Act [38], and cleared by the North West Regional Delegation of Public Health (Reference N°: 95/ATT/NWR//RDPH/ BRIGAD). Only respondents who verbally consented to the study, were allowed to participate by filling and submitting the anonymous questionnaire.

## Results

### Characteristics of study population

Six hundred and ninety (690) persons were logged into our Microsoft Excel sheet, from whom 25 were dropped as outliers for age, BMI, and prolonged hospitalisation. Of the 665 persons included in this study, 2,126 household residents were counted; 1,751 (82.4%) were persons 0 – 59 years old and 375 (17.6%) were persons $\geq 60$ years old. Four hundred and thirty-seven (52.2%) persons were females and 318 (47.8%) were males (Table 1).

The mean age of the participants was 34 years (SD 9.0, range 20 – 64). Three hundred and ninety (58.6%) were singles, 308 (45.9%) were of tertiary educational status, and 268 (40.3%) were skilled workers.

### Clinical profile of respondents

**Pre- and post- COVID-19 diagnosis/treatment symptoms.** Three hundred and ten (46.6%), 289 (43.5%), and 214 (32.2%) of the respondents presented with cough, sore throat, and tiredness, respectively (Fig 1) for a mean duration of 2.30 days (SD 2.46, range 0 – 9 days) prior to diagnosis and treatment. Fatigue/Tiredness, dry/unproductive cough, and shortness of breath were experienced for a mean duration of 3.65 days (SD 3.17, range 0 – 8) after drug administration.

Table 1. Characteristics of study participants.

| General characteristic | Subclass | Count (%) |
|---|---|---|
| **Age groups (in years)** | ≤ 40 | 510 (76.7) |
| | > 40 | 155 (23.3) |
| | Mean age ($\bar{x}$ ± SD) | 34.00 ± 9.00 |
| **Sex** | Female | 437 (52.2) |
| | Male | 318 (47.8) |
| **BMI (Kg/m$^2$)** | Eutrophic | 256 (38.5) |
| | Overweight | 233 (35.0) |
| | Obese | 176 (26.5) |
| | Mean BMI ($\bar{x}$ ± SD) | 27.00 ± 4.00 |
| **Marital status** | Not married | 390 (58.6) |
| | Married | 275 (41.4) |
| **Education** | Primary | 142 (21.4) |
| | Secondary | 218 (32.8) |
| | Tertiary | 305 (45.9) |
| **Occupation** | Business/Private sector | 118 (17.7) |
| | Student | 97 (14.6) |
| | Unemployed | 118 (17.7) |
| | Unskilled worker | 64 (9.6) |
| | Skilled worker | 268 (40.3) |
| **Household size** | 1 – 4 | 546 (82.1) |
| | 5 – 9 | 119 (17.9) |
| | Mean household size ($\bar{x}$ ± SD) | 3.00 ± 1.00 |
| **Number of sleeping rooms** | One | 209 (31.4) |
| | Two | 365 (54.9) |
| | More than two (3 - 5) | 91 (13.7) |
| | Mean number of rooms ($\bar{x}$ ± SD) | 1.00 ± 0.00 |
| **Comorbid condition** | Gastritis | 202 (30.4) |
| | Previously operated | 89 (13.4) |
| | High blood pressure | 108 (16.2) |
| | Hepatitis/Liver disease | 95 (14.3) |
| | Diabetes | 50 (7.5) |
| **Number of comorbidities** | None | 202 (30.4) |
| | At least one | 384 (57.7) |
| | At most two | 77 (11.7) |

BMI; Body Mass Index [Eutrophic (18.5 ≤ BMI ≥ 24.9), Overweight (25.0 ≤ BMI ≥ 29.9), Obese (BMI ≥ 30.0)], SD; Standard Deviation.

Two hundred and sixty-eight (40.3%) and 316 (47.5%) of the respondents, respectively, manifested with no symptoms of COVID-19 prior to testing positive and after treatment (Fig 1).

**Severity and comorbidities.** The severity of pre- and post-COVID-19 diagnosis and treatment are presented in Fig 2. The most common comorbidities were gastritis 202 (30.4%), high blood pressure 108 (16.2%), and hepatitis and/or other liver diseases 95 (14.3%), with 202 (30.4%) having no comorbidity and 77 (11.7%) having at most two comorbidities (Table 1).

**Duration of hospitalisation/quarantine.** The mean duration of hospitalisation and quarantine periods were, respectively, 2.95 days (SD 7.20, range 0 – 24) and 6.84 days (SD 7.00,

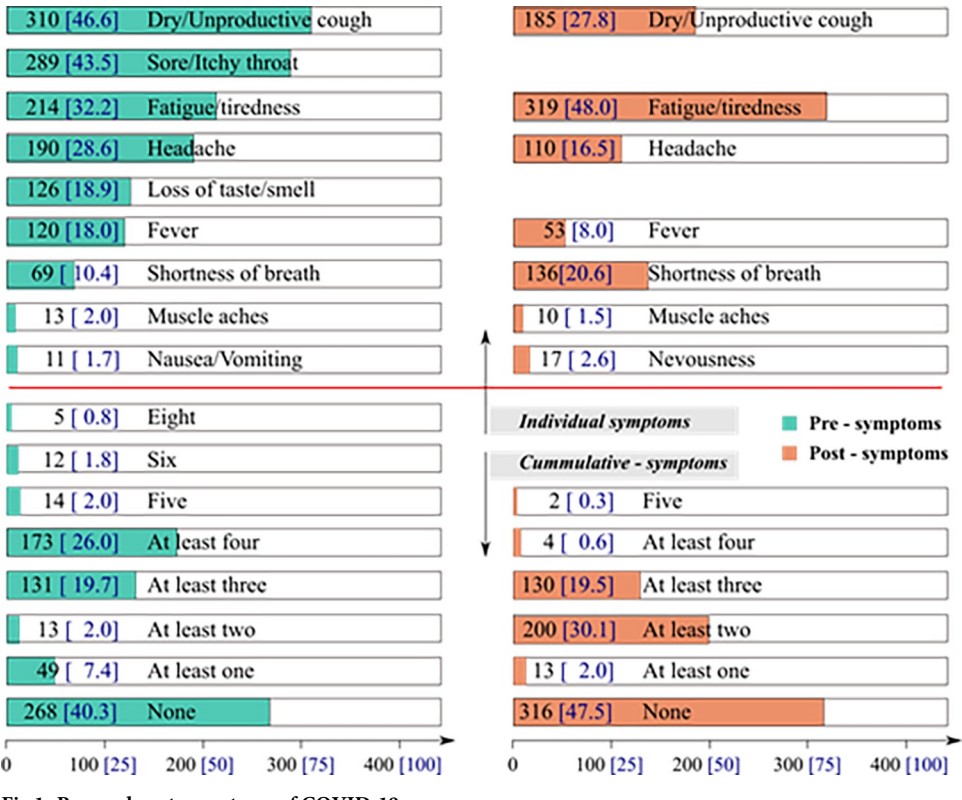

**Fig 1. Pre- and post-symptoms of COVID-19.**

range 0 – 28). Male respondents stayed longer in the hospital and in quarantine than females (Table 2).

**Care, treatment and vaccination hesitancy.** Patients received either the Ministry of Health's (MOH) protocol of oral hydroxychloroquine, paracetamol, vitamin C, zinc, and

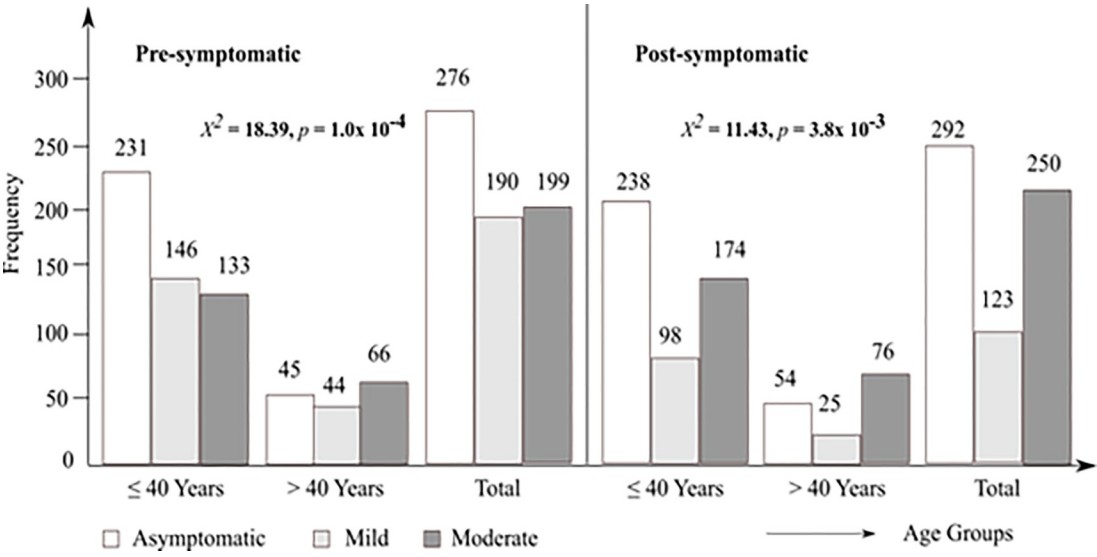

**Fig 2. Severity grading for pre- and post-COVID-19 amongst age groups.**

**Table 2. Association of hospitalisation/quarantination with respondents' characteristics; ANOVA.**

| Variable | Subclass | Hospitalisation (*in days*) | | Quarantination (*in days*) | |
|---|---|---|---|---|---|
| | | $\bar{x} \pm$ SD | F ($p$ – value) | $\bar{x} \pm$ SD | F ($p$ – value) |
| **Age (in years)** | $\leq 40$ | $2.92 \pm 7.20$ | $2.0\times10^{-2}$ ($8.6\times10^{-1}$) | $6.70 \pm 7.03$ | $1.1\times10^{\circ}$ ($3.0\times10^{-1}$) |
| | $> 40$ | $3.02 \pm 7.12$ | | $7.35 \pm 6.74$ | |
| **Sex** | Female | $2.62 \pm 6.84$ | $1.5\times10^{-1}$ ($2.2\times10^{-1}$) | $6.81 \pm 6.93$ | $1.2\times10^{-2}$ ($9.1\times10^{-1}$) |
| | Male | $3.30 \pm 7.48$ | | $6.87 \pm 7.02$ | |
| **BMI (kg/m$^2$)** | Eutrophic | $2.52 \pm 6.79$ | $2.8$ ($2.4\times10^{-1}$) | $6.58 \pm 6.65$ | $3.0\times10^{-1}$ (**$5.3\times10^{-2}$**) |
| | Overweight | $3.30 \pm 7.56$ | | $7.05 \pm 6.71$ | |
| | Obese | $3.13 \pm 7.14$ | | $6.95 \pm 7.74$ | |
| | Total | $2.95 \pm 7.20$ | | $6.84 \pm 7.00$ | |

**Bold** numbers are significant $p$ – values. BMI; Body Mass Index [Eutrophic ($18.5 \leq$ BMI $\geq 24.9$), Overweight ($25.0 \leq$ BMI $\geq 29.9$), Obese (BMI $\geq 30.0$)]

azithromycin or the local remedy ADSAK/ELEXIR COVID or both. Depending on the severity and comorbidities, some patients received or added extra doses to the prescribed doses.

Five hundred and seventy-three (86.2%) of the respondents were prescribed and administered the hydroxychloroqine/Azithromycin/Paracetamol/Zinc/Vitamin C regimen, 25 (3.8%) were simply advised to go home, and 230 (34.6%) opted to supplement the MOH's regimen with the local remedy ADSAK/ELIXIR COVID (Table 3).

Only 165 (24.8%) of the respondents admitted that they will accept the vaccine when it is feasible, thus yielding a COVID-19 hesitancy rate of 74% (Fig 3). From multinomial regression analysis, the odds for denying COVID-19 vaccine was higher amongst the unemployed (OR; 1.5, 95%C.I; 0.8 – 3.0), unskilled workers (OR; 1.4, 95%C.I; 0.7 – 3.0), and those residing in Bamenda and Bafoussam [(OR; 1.3, 95%C.I; 0.6 – 2.5) vs (OR; 1.2, 95%C.I; 0.6 – 2.4)], when compared with their counterparts (S3 Table in S1 File).

## Ownership/Utilisation and sources of face masks

**Ownership/Utilisation of face masks.** Of the 665 respondents sampled, 155 (23.3%) were in households wherein all residents owned FMs (Fig 4).

Two hundred and seventy-one (40.8%) of the respondents correctly wore their FMs to cover the nose, mouth, and chin, such so as to be able to breathe, while 313 (47.1%) adequately/effectively used FMs for $\geq 6$ days/week (Fig 4).

**Factors associated with ownership/utilisation of face masks.** Multinomial logistic regression was used to test if respondent characteristics were significantly associated with face mask ownership/utilisation. The results indicated that the odds for the ownership of FMs by

**Table 3. Treatment regimens for respondents.**

| Treatment | Frequency | Percent |
|---|---|---|
| Counselled to go home for self-quarantine | 25 | 3.8 |
| Hydroxychloroqine/Azithromycin/Paracetamol/Zinc/Vitamin C | 573 | 86.2 |
| ADSAK/ELIXIR COVID | 230 | 34.6 |
| **Cumulative treatment** | | |
| None | 81 | 12.2 |
| At least one treatment regimen | 340 | 51.1 |
| Both treatment regimens | 244 | 36.7 |
| Supplemented medication | 260 | 39.1 |

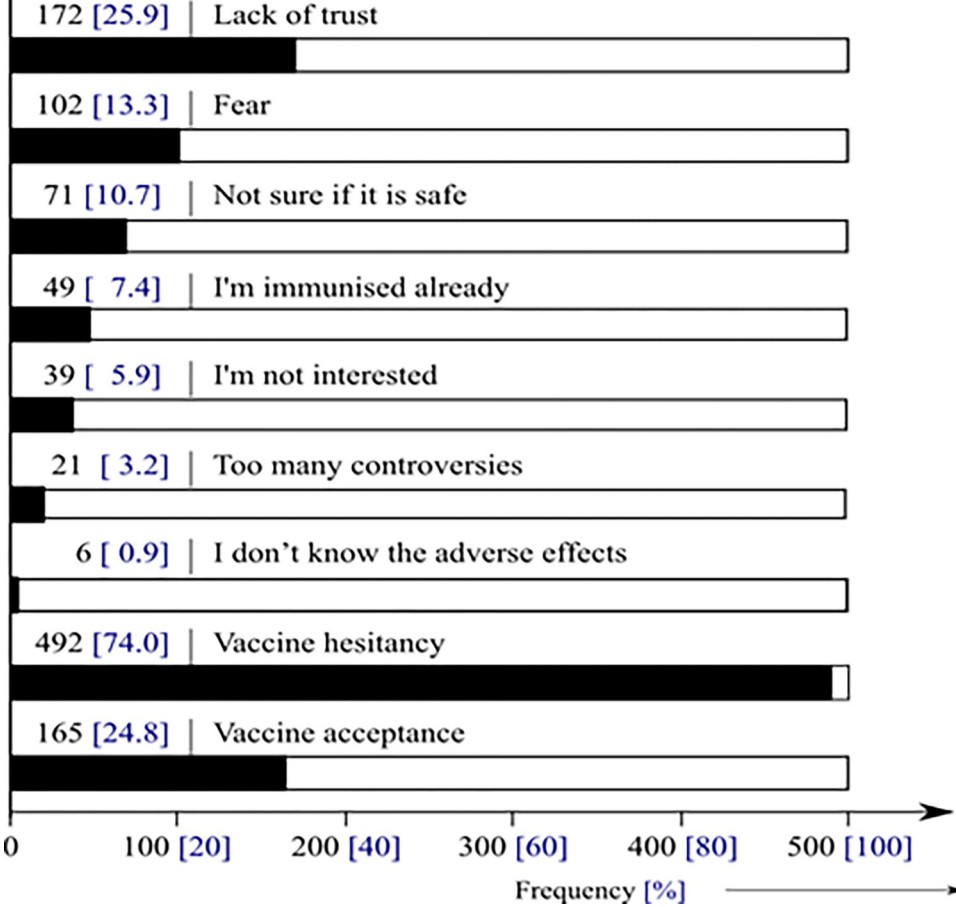

**Fig 3. COVID-19 Vaccine hesitancy and reasons for hesitancy.**

the entire household were higher amongst the male sex (OR; 1.1, 95%C.I; 0.7 – 1.5), married persons (OR; 1.3, 95%C.I; 0.8 – 2.0), primary school leavers (OR; 1.3, 95%C.I; 0.6 – 2.6), students as well as business operators [(OR; 1.6, 95%C.I; 0.9 – 3.1) vs (OR; 1.3, 95%C.I; 0.7 – 2.1)] and Bamenda as well as Buea [($p$; $<1.0\times10^{-3}$, OR; 7.6, 95%C.I; 3.8 – 15.5) vs ($p$; $<1.0\times10^{-3}$, OR; 14.6, 95%C.I; 6.5 – 33.1)] (Table 4).

Respondents with the primary level of education ($p$; $9.5\times10^{-1}$, OR; 1.1, 95% C.I; 0.6 – 1.8), were more likely to correctly use FMs than those with the tertiary educational status. More females had FMs than males and were less likely to correctly wear them compared to males ($p$; $1.2\times10^{-1}$, OR; 0.8, 95% C.I; 0.6 – 1.1) (Table 4).

The reasons advanced for the irregular use of FMs from respondents' perspective were: 'it is hard to be consistent' (39.7%), 'I wear them whenever I am going outdoors' (28.7%), and 'it is boring/uncomfortable' (27.2%) (Fig 5).

**Source of face masks.** Respondents either purchased FMs, 58.8% or obtained them freely from the office/workplace 35.8%, and a gift from a relationship 35.2%, or self-made 17.3% (Table 5).

## Knowledge of COVID-19 symptoms and household preventive measures against it

**Knowledge of COVID-19 symptoms.** Of the 10 symptoms anticipated, 21 (3.2%) of the respondents indicated ignorance, while 258 (38.8%) were informed on all 10 (S1 Table in

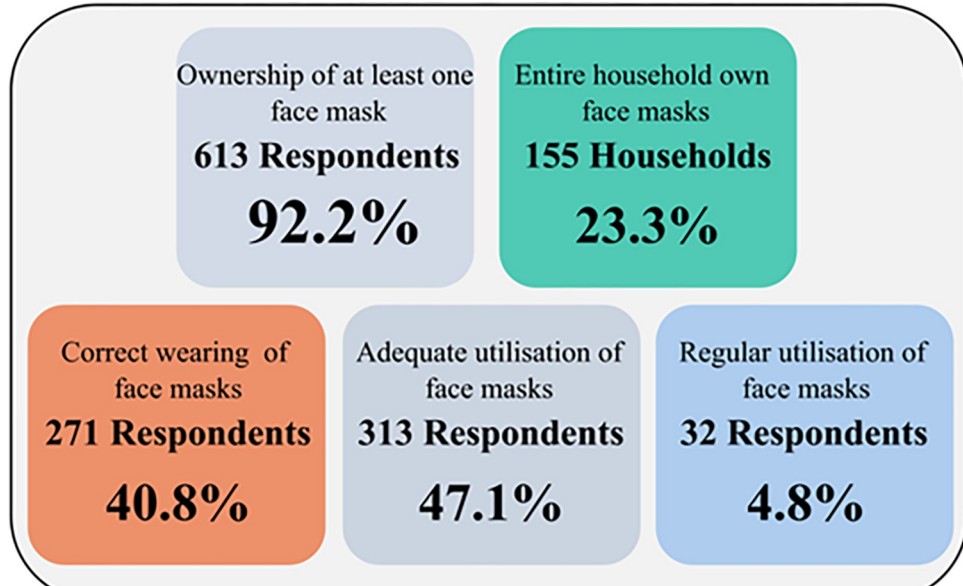

**Fig 4. Ownership/utilisation of face masks.**

S1 File). Four hundred and forty (66.2%) of the respondents have very good knowledge of COVID-19 symptoms, with a knowledge score of $\geq$80%. Participants, 40 years or less ($p$; $1.7\times10^{-2}$, OR; 1.7, 95% C.I; 1.1 - 2.6), unskilled workers as well as business operators [(OR; 1.3, 95% C.I; 0.6 - 2.7) vs (OR; 1.2, 95% C.I; 0.7 - 2.0)], and those residing in Buea as well as Douala [(OR; 1.3, 95% C.I; 0.6 - 2.9) vs (OR; 1.1, 95% C.I; 0.7 - 1.6)], higher odds to have very good knowledge of COVID-19 symptoms when compared with their counterparts (Table 6).

**Household preventive measures of COVID-19.**  Of the five preventive measures, 570 (85.7%) of the respondents pointed out to regular hand washing, 75 (11.3%) made no mention of any of the measures, while 110 (16.5%) made mention of four of the measures (S2 Table in S1 File).

Two hundred and twenty-two (8.4%) implemented moderate ($\geq$50%) preventive measures against COVID-19. The odds of implementing COVID-19 preventive measures was higher amongst those who were 40 years or less (OR; 1.4, 95%C.I: 0.9 - 2.2), students ($p$; $8\times10^{-3}$, OR; 0.5, 95%C.I: 0.3 - 0.8), and those residing in Bamenda ($p$; $1\times10^{-2}$, OR; 0.4, 95%C.I: 0.2 - 0.8), when compare with those older than 40 years, skilled workers, and those residing in Yaoundé respectively (Table 6).

## Discussion

Our study adds to the description of COVID-19 in Cameroon where there is a large gap in the literature. Our study revealed that; pre-/post symptoms were dry cough and tiredness, with sore throat and headache occurring more before diagnosis/treatment while shortness of breath persisted after treatment. Gastritis, high blood pressure, and hepatitis were the underlying comorbidities of COVID-19 amongst respondents. Majority of the respondents owned FMs, few households owned FMs, while the use of FMs was generally low in all aspects. Many respondents feared COVID-19 vaccines leading to a very high VH rate.

### Pre- and post – COVID-19 diagnosis/treatment symptoms

Cough, tiredness, and headache occurred in patients before and after treatment. A study from Ethiopia proved that these symptoms were confirmed amongst COVID-19 cases, with other

**Table 4. Association of respondents' characteristics with face mask ownership/utilisation.**

| Characteristic | Ownership of FM (n = 613) | | | Entire household own FM (n = 155) | | |
|---|---|---|---|---|---|---|
| | n (%) | p-value | OR (95% C.I) | n (%) | p-value | OR (95% C.I) |
| **Age groups (in years)** | | | | | | |
| $\leq 40/> 40$ | 472/141 | $2.6 \times 10^{-1}$ | 1.5 (0.7 – 3.1) | 110/45 | $8.7 \times 10^{-1}$ | 1.0 (0.6 – 1.8) |
| **Sex** | | | | | | |
| Male/Female | 286/327 | $\mathbf{4.5 \times 10^{-2}}$ | 0.5 (0.3 – 1.0) | 76/79 | $9.1 \times 10^{-1}$ | 1.1 (0.7 – 1.5) |
| **Marital status** | | | | | | |
| Married/Not married | 253/360 | $2.9 \times 10^{-1}$ | 0.7 (0.4 – 1.3) | 68/87 | $2.4 \times 10^{-1}$ | 1.3 (0.8 – 2.0) |
| **Education** | | | | | | |
| Secondary/Tertiary | 198/283 | $4.1 \times 10^{-1}$ | 0.7 (0.4 – 1.5) | 42/89 | $9.6 \times 10^{-1}$ | 1.0 (0.5 – 1.6) |
| Primary/Tertiary | 132 (21.5) | $9.7 \times 10^{-1}$ | 1.1 (0.4 – 3.0) | 24 (15.5) | $5.6 \times 10^{-1}$ | 1.3 (0.6 – 2.6) |
| **Occupation** | | | | | | |
| Student/Skilled worker | 87/244 | $6.2 \times 10^{-1}$ | 0.8 (0.3 – 1.9) | 27/69 | $1.3 \times 10^{-1}$ | 1.6 (0.9 – 3.1) |
| Unemployed/Skilled worker | 108 (17.6) | $7.0 \times 10^{-1}$ | 0.8 (0.3 – 2.3) | 16 (10.3) | $4.3 \times 10^{-1}$ | 0.7 (0.3 – 1.6) |
| Unskilled/Skilled worker | 61 (9.9) | $4.9 \times 10^{-1}$ | 1.6 (0.4 – 6.5) | 6 (3.9) | $1.8 \times 10^{-1}$ | 0.5 (0.2 – 1.4) |
| Business operator/Skilled worker | 113 (18.4) | $2.1 \times 10^{-1}$ | 2.0 (0.7 – 4.4) | 37 (23.9) | $3.9 \times 10^{-1}$ | 1.3 (0.7 – 2.9) |
| **Household size** | | | | | | |
| 5 – 9/1 – 4 | 115/498 | $\mathbf{4.1 \times 10^{-2}}$ | **3.2 (1.1 – 9.6)** | 27/128 | $7.0 \times 10^{-2}$ | 0.6 (0.3 – 1.0) |
| **Residence** | | | | | | |
| Bamenda/Yaoundé | - | - | - | 31/44 | $\mathbf{<1.0 \times 10^{-3}}$ | **7.6 (3.8 – 15.5)** |
| Bafoussam/Yaoundé | - | - | - | 8 (5.2) | $6.3 \times 10^{-1}$ | 0.8 (0.3 – 1.9) |
| Buea/Yaoundé | - | - | - | 35 (22.6) | $\mathbf{<1.0 \times 10^{-3}}$ | **14.6 (6.5 – 33.1)** |
| Douala/Yaoundé | - | - | - | 37 (23.9) | $2.4 \times 10^{-1}$ | 0.7 (0.5 – 1.2) |
| | **Correct wearing of FM (n = 271)** | | | **Adequate utilisation of FM (n 313)** | | |
| **Age groups (in years)** | | | | | | |
| $\leq 40/> 40$ | 209/62 | $9.9 \times 10^{-1}$ | 1.0 (0.7 – 1.5) | 250/63 | $\mathbf{2.2 \times 10^{-3}}$ | 1.9 (1.3 – 2.9) |
| **Sex** | | | | | | |
| Male/Female | 119/152 | $1.2 \times 10^{-1}$ | 0.8 (0.6 – 1.1) | 152/161 | $7.1 \times 10^{-1}$ | **1.1** (0.8 – 1.4) |
| **Marital status** | | | | | | |
| Married/Not married | 103/168 | $\mathbf{1.3 \times 10^{-2}}$ | 0.6 (0.5 – 1.0) | 123/190 | $6.1 \times 10^{-2}$ | 0.7 (0.5 – 1.0) |
| **Education** | | | | | | |
| Secondary/Tertiary | 74/133 | $\mathbf{4.1 \times 10^{-2}}$ | 0.6 (0.4 – 1.0) | 99/152 | $7.9 \times 10^{-1}$ | 0.9 (0.6 – 1.4) |
| Primary/Tertiary | 64 (23.6) | $9.5 \times 10^{-1}$ | 1.1 (0.6 – 1.8) | 62 (19.8) | $7.4 \times 10^{-1}$ | 0.9 (0.5 – 1.6) |
| **Occupation** | | | | | | |
| Student/Skilled worker | 35/114 | $5.5 \times 10^{-1}$ | 0.8 (0.5 – 1.4) | 39/149 | $\mathbf{5.0 \times 10^{-4}}$ | 0.4 (0.2 – 0.6) |
| Unemployed/Skilled worker | 55 (20.3) | $6.1 \times 10^{-1}$ | 1.2 (0.6 – 2.1) | 55 (17.6) | $8.2 \times 10^{-2}$ | 0.5 (0.3 – 1.1) |
| Unskilled/Skilled worker | 25 (9.2) | $9.4 \times 10^{-1}$ | 1.0 (0.5 – 1.9) | 29 (9.3) | $8.6 \times 10^{-2}$ | 0.5 (0.3 – 1.1) |
| Business operator/Skilled worker | 42 (15.5) | $2.9 \times 10^{-1}$ | 0.8 (0.5 – 1.2) | 41 (13.1) | $\mathbf{1.0 \times 10^{-4}}$ | 0.3 (0.2 – 0.6) |
| **Household size** | | | | | | |
| 5 – 9/1 – 4 | 49/222 | $6.2 \times 10^{-1}$ | 1.1 (0.7 – 1.7) | 60/253 | $8.7 \times 10^{-2}$ | 1.4 (0.9 – 2.3) |
| **Residence** | | | | | | |
| Bamenda/Yaoundé | 15/95 | $2.4 \times 10^{-1}$ | 0.6 (0.3 – 1.3) | 19/121 | $5.9 \times 10^{-2}$ | 0.5 (0.3 – 1.0) |
| Bafoussam/Yaoundé | 30 (1.1) | $\mathbf{1.1 \times 10^{-2}}$ | 2.3 (1.2 – 4.6) | 26 (8.3) | $7.1 \times 10^{-1}$ | 0.9 (0.4 – 1.7) |
| Buea/Yaoundé | 19 (7.0) | $9.3 \times 10^{-1}$ | 1.0 (0.5 – 1.9) | 23 (7.4) | $9.7 \times 10^{-1}$ | 1.0 (0.5 – 2.0) |
| Douala/Yaoundé | 112 (41.3) | $6.1 \times 10^{-1}$ | 1.1 (0.8 – 1.6) | 124 (39.6) | $4.0 \times 10^{-1}$ | 0.8 (0.6 – 1.2) |

OR = Odds Ratio; C.I. = Confidence Interval; **Boldface** numbers indicate significant p values.

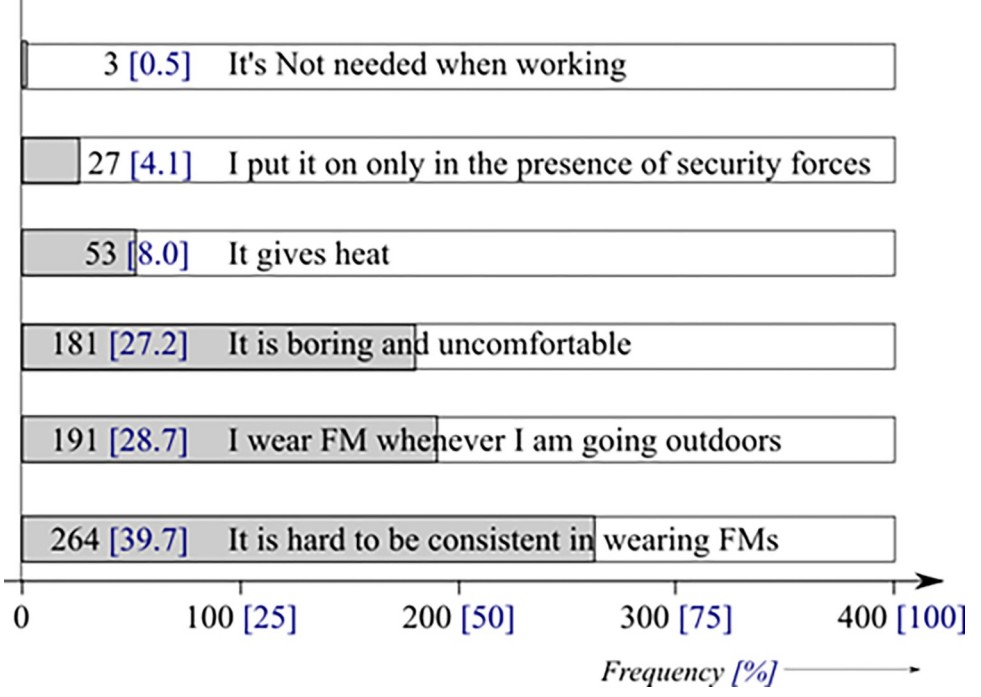

**Fig 5. Reasons for irregular use of face masks.**

common symptoms being sore throat [27]. Fever, fatigue, and cough, however, remain the most common symptoms in many cohort studies [28, 39, 40].

Disease severity ranged from asymptomatic (41.5% vs 43.9%) to mild (28.6% vs 18.5%) to moderate (29.9% vs 37.6%) before and after treatment. This was similar to findings reported in Ethiopia [27] but different from the findings of another study in Cameroon [28]. Nearly two-thirds of the patients suffered from at least one comorbidity (57.7%), of which gastritis was the most common (30.4%). This was similar to the 29% reported in Ethiopia [27], and different from the findings of another study [41], in which it was asserted that the most common comorbidities were obesity, diabetes and hypertension. These differences may be as a result of differences in study population and study design.

The mean duration of hospitalisation and quarantine period were, respectively, 2.95 days (SD 7.20, range 0 – 24) and 6.84 days (SD 7.00, range 0 – 28). This was lower compared with mean hospitalisation of five days reported in Cameroon [28] and the 13.5 – 42 days (SD, 9.7) reported in studies elsewhere [27, 39, 40]. Recovery, however, depends on the severity of the infection as well as comorbidities. Severity seems to have dropped with time, thus leading to both little hospitalisation and quarantine periods.

Five hundred and seventy-three (86.2%) of the respondents were prescribed and administered the hydroxychloroqine/Azithromycin/Paracetamol/Zinc/Vitamin C regimen, 3.8% were counselled to go home. In a similar study, all patients received a treatment protocol with oral

**Table 5. Sources of face masks.**

| Source of face masks | Frequency | Percent |
|---|---|---|
| A gift from a relationship | 234 | 35.2 |
| Purchased from the shop/Pharmacy | 238 | 35.8 |
| From the office/Workplace | 391 | 58.8 |
| Self-made/Tailor | 115 | 17.3 |

**Table 6. Association of respondent's characteristics with knowledge of COVID-19 symptoms/household preventive measures of COVID-19.**

| Respondent's Characteristic | VG Knowledge of symptoms (n = 440) | | | Moderate HH preventive measures (n = 222) | | |
|---|---|---|---|---|---|---|
| **Age groups (in years)** | *n* (%) | *p*-value | OR (95% C.I) | *n* (%) | *p*-value | OR (95% C.I) |
| $\leq$ 40/> 40 | 344/96 | **$1.7 \times 10^{-2}$** | 1.7 (1.1 - 2.6) | 175/47 | $1.2 \times 10^{-1}$ | 1.4 (0.9 - 2.2) |
| **Sex** | | | | | | |
| Male/Female | 200/240 | $3.6 \times 10^{-1}$ | 0.8 (0.6 - 1.2) | 96/126 | $1.9 \times 10^{-1}$ | 0.8 (0.6 - 1.1) |
| **Marital status** | | | | | | |
| Married/Not married | 185/255 | $8.0 \times 10^{-1}$ | 0.9 (0.6 - 1.4) | 89/133 | $2.7 \times 10^{-1}$ | 0.8 (0.6 - 1.1) |
| **Education** | | | | | | |
| Secondary/Tertiary | 142/205 | $5.0 \times 10^{-1}$ | 0.9 (0.6 - 1.3) | 72/104 | $5.8 \times 10^{-1}$ | 0.8 (0.6 - 1.3) |
| Primary/Tertiary | 93 (21.1) | $1.9 \times 10^{-1}$ | 0.6 (0.4 - 1.2) | 46 (20.7) | $4.9 \times 10^{-1}$ | 0.8 (0.4 - 1.5) |
| **Occupation** | | | | | | |
| Student/Skilled worker | 53/182 | **$3.8 \times 10^{-3}$** | 0.5 (0.3 - 0.8) | 24/96 | **$8.0 \times 10^{-3}$** | 0.5 (0.3 - 0.8) |
| Unemployed/Skilled worker | 75 (17.1) | $6.0 \times 10^{-1}$ | 0.8 (0.5 - 1.6) | 36 (16.2) | $2.0 \times 10^{-1}$ | 0.7 (0.4 - 1.2) |
| Unskilled/Skilled worker | 47 (10.7) | $4.8 \times 10^{-1}$ | 1.3 (0.6 - 2.7) | 24 (10.8) | $8.4 \times 10^{-1}$ | 0.9 (0.5 - 1.8) |
| Business operator/Skilled worker | 83 (18.9) | $5.1 \times 10^{-1}$ | 1.2 (0.7 - 2.0) | 42 (18.9) | $8.5 \times 10^{-1}$ | 0.9 (0.6 - 1.5) |
| **Household size** | | | | | | |
| 5 – 9/1 – 4 | 76/364 | $6.0 \times 10^{-1}$ | 0.9 (0.6 - 1.4) | 40/182 | $5.1 \times 10^{-1}$ | 1.7 (0.7 - 1.8) |
| **Residence** | | | | | | |
| Bamenda/Yaoundé | 21/167 | **$4.0 \times 10^{-4}$** | 0.3 (0.2 - 0.6) | 9/87 | **$1.0 \times 10^{-2}$** | 0.4 (0.2 - 0.8) |
| Bafoussam/Yaoundé | 31 (7.0) | **$4.5 \times 10^{-2}$** | 0.5 (0.3 - 1.0) | 14 (6.3) | $7.8 \times 10^{-2}$ | 0.5 (0.3 - 1.1) |
| Buea/Yaoundé | 35 (7.9) | $4.6 \times 10^{-1}$ | 1.3 (0.6 - 2.9) | 13 (5.9) | $1.7 \times 10^{-1}$ | 0.6 (0.3 - 1.2) |
| Douala/Yaoundé | 186 (42.3) | $6.8 \times 10^{-1}$ | 1.1 (0.7 - 1.6) | 99 (44.6) | $7.4 \times 10^{-1}$ | 1.1 (0.7 - 1.6) |

OR = Odds Ratio; C.I. = Confidence Interval; VG = Very Good; HH = Household; **Boldface** numbers indicate significant *p* values.

chloroquine, paracetamol, vitamin C, zinc, amoxicillin combined with clavulanic acid, and azithromycin. Further, some patients received anticoagulants, corticosteroids, or intravenous antibiotics, with 15% of confirmed cases undergoing non-invasive ventilation [27].

## COVID-19 vaccine hesitancy/acceptance

In this study, vaccine hesitancy (VH) was 74.0% (vaccine acceptance of 24.8%) for a variety of reasons; lack of trust and fear. This was similar to the 71 - 84.6% reported amongst Cameroonians [26, 27], as well as the 74.3% COVID-19 vaccine rejection in Bosnia and Herzegovina [42], and higher than the 62.4 – 62.6% reported elsewhere in the Arab world and Jordan [43, 44]. The reasons for high VH were lack of trust in government's decisions [26], people's reluctance to get the vaccine, lack of dedicated health personnel to vaccination [16], the role of social media environments, perception of pharmaceutical industry, reliability and source of vaccine [25, 45]. Generally, there is a high rate (57.4%) of COVID-19 fear as reported in Cameroon [12], which has however dropped with time. The fear of COVID-19 has dropped for the following undocumented reasons; fear of vaccines, the assumption that COVID-19 is a simple flu, and lack of enough confidence in the vaccine development process [46]. Vaccine acceptance rates are 94.4 – 97% in Asia, 62.4% in the Arab world, 23.6% in Kuwait, 28.4% in Jordan, and 53.7 – 58.9% in Europe, and rates of 27.7% in Democratic Republic of Congo [43, 47].

## Face mask ownership/utilisation

In this study, 155 (23.3%) respondents were in households wherein all residents owned FMs, 613 (92.2%) owned FMs, 271 (40.8%) correctly wore them, and 313 (47.1%) adequately used

FMs for $\geq$ 6 days/week. Adherence to the correct and effective use of FMs in this study was higher compared with 1.4% reported amongst Polish Health Care Workers [48]. The differences may be due to the fact that our study had a variety of occupations, while that reported in Poland was amongst Health Care Workers only.

In another study, respondents used FMs in various occasions: 95.3% for protection against COVID-19, 90.2% used FMs in public, 53% used it when entering restricted places, 45.5% when with a suspected case and 30.7% used a mask due to fear of arrest/punishment [49]. In the course of time, the use of FM too is decreasing as many, simply say COVID-19 does not exist.

Four hundred and forty (66.2%) of the respondents have very good knowledge of COVID-19 symptoms. In another study, 58.6% had moderate knowledge about COVID-19, whereas 37.2% had good knowledge [7]. The knowledge level in this study is higher compared to the 21.9% reported in Buea – Cameroon [29] and lower compared to the 68.2% reported in a multinational study [50].

### COVID-19 preventive measures

Of the five preventive measures, 570 (85.7%) of the respondents pointed out to regular hand washing, 75 (11.3%) made no mention of any of the measures, while 110 (16.5%) made mention of four of the measures. These findings were different from the adherence to barrier measures reported in Cameroon[12].

Fundamental to assessing pre-/post- COVID-19 symptoms, and ownership and utilisation rates of FMs is obtaining epidemiological data from the communities. These findings underline the need for continuous intervention programmes to enhance the prevention of the spread of COVID-19. Regular health education on COVID-19 vaccinations, and the practice of preventive measures should be encouraged. Stakeholders should try to curb vaccine hesitation by involving local scientists in the development or manufacturing process, and by dealing with conspiracies.

## Strengths and limitations

### Strengths

Field data were obtained online and from workplace/door-to-door/OPDs. The quality of the data collected was assured through pretesting of questionnaires to minimize bias as well as errors.

### Limitations

This was a cross-sectional study, representing the snapshot of the population within the study period and does not show cause and effect since the predictor and outcome variables were measured at the same time. Data was collected through anonymous self-reporting via partly online and partly workplace/door-to-door/OPD and thus there is a possibility of various types of bias; selection/double selection bias, and recall bias. Such biases can also affect some of the responses and subsequently the results of the study. Other limitations to this study were that online respondents might not have met the inclusion criteria, FMs were not categorised into surgical, air filtering respirators or simple cloth masks.

## Conclusion

In this study, we described the symptoms and severity of COVID-19 infection as well as length of hospital stay/quarantine, face mask utilisation and vaccine hesitation/acceptance amongst

persons who have recovered from COVID - 19. There was a variety of symptoms with unproductive cough and fatigue as the most prevalent symptoms occurring before and after treatment. The majority of patients had no comorbidity and the commonest comorbidities were gastritis and high blood pressure. Despite the prescription of various treatment regimens, the length of stay in the hospital or quarantine was long.

Thus, the treatment of COVID-19 does not mean immediate recovery as the mild to moderate grade severity persists. Face mask availability and ownership does not mean higher utilisation. This study advocates for an intensification of COVID-19 preventive practices, as well as elaborate education on the importance of vaccination and preparation for future disease outbreaks.

## Supporting information

**S1 Checklist.**
(DOC)

**S1 Appendix. Survey questionnaire.**
(PDF)

**S1 File. Supplementary tables.**
(DOCX)

**S2 File.**
(DOCX)

**S1 Data.**
(XLSX)

## Acknowledgments

We are thankful to the respondents who participated in this survey and to the survey teams; especially those of Douala and Yaoundé. We wish to greatly acknowledge the Central Africa Network for Tuberculosis, HIV/AIDS and Malaria (CANTAM), for inputs on data analysis and research methods.

## Author Contributions

**Conceptualization:** Frederick Nchang Cho, Yayah Emerencia Ngah.

**Data curation:** Yayah Emerencia Ngah, Patrick Kofon Jokwi.

**Formal analysis:** Frederick Nchang Cho.

**Investigation:** Frederick Nchang Cho, Yayah Emerencia Ngah, Patrick Kofon Jokwi, Valmie Ngassam Folefac.

**Methodology:** Frederick Nchang Cho, Yayah Emerencia Ngah, Peter Canisius Kuku Elad, Valmie Ngassam Folefac.

**Project administration:** Ismaila Esa, Paulette Ngum Fru.

**Resources:** Andrew N. Tassang, Celestina Neh Fru, Ismaila Esa.

**Supervision:** Yayah Emerencia Ngah, Andrew N. Tassang, Paulette Ngum Fru.

**Validation:** Frederick Nchang Cho, Andrew N. Tassang, Celestina Neh Fru, Peter Canisius Kuku Elad, Patrick Kofon Jokwi, Valmie Ngassam Folefac, Ismaila Esa, Paulette Ngum Fru.

**Writing – original draft:** Frederick Nchang Cho, Yayah Emerencia Ngah, Peter Canisius Kuku Elad, Patrick Kofon Jokwi.

**Writing – review & editing:** Frederick Nchang Cho, Yayah Emerencia Ngah, Peter Canisius Kuku Elad, Patrick Kofon Jokwi.

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
