## [Decision Letter · Decision Letter 0]

19 Sep 2022

PONE-D-22-15424Pre- and Post-COVID-19 Diagnosis/Treatment Symptoms, Face mask Ownership/Utilisation, COVID-19 Preventive Measures, and COVID-19 Vaccine Hesitancy Amongst Patients Recovering from the COVID-19 infection in Cameroon: A Cross-Sectional StudyPLOS ONE

Dear,

Thank you for submitting your manuscript to PLOS ONE. After careful consideration, we feel that it has merit but does not fully meet PLOS ONE’s publication criteria as it currently stands. Therefore, we invite you to submit a revised version of the manuscript that addresses the points raised during the review process. The reviewer comment has been given below.

We look forward to receiving your revised manuscript.

Kind regards,

Muhammad Shahzad Aslam, Ph.D.,M.Phil., Pharm-D

Academic Editor

PLOS ONE

Journal Requirements:

2. Please provide additional details regarding participant consent. In the ethics statement in the Methods and online submission information, please ensure that you have specified what type you obtained (for instance, written or verbal, and if verbal, how it was documented and witnessed). If your study included minors, state whether you obtained consent from parents or guardians. If the need for consent was waived by the ethics committee, please include this information

Reviewers' comments:

Reviewer's Responses to Questions

**Comments to the Author**

1. Is the manuscript technically sound, and do the data support the conclusions?

Reviewer #1: Yes

Reviewer #2: Partly

2. Has the statistical analysis been performed appropriately and rigorously? 

Reviewer #1: Yes

Reviewer #2: Yes

3. Have the authors made all data underlying the findings in their manuscript fully available?

Reviewer #1: No

Reviewer #2: Yes

4. Is the manuscript presented in an intelligible fashion and written in standard English?

Reviewer #1: Yes

Reviewer #2: No

5. Review Comments to the Author

Reviewer #1: Line 37, Your study period is April - July, and August - October 2021 but you however present 2022 data in lines 58 and 69. Kindly review

In line 99 where you mention the aim of the study, you don't talk about Diagnosis/treatment symptoms but emphasize pre and post COVID 19 Symptoms. It is important to be uniform in the presentation of information to ease comprehension.

The introduction section does not cleary reveal the rationale of your study. Do well to bring our the rationale of this study in the introduction section

Your study population and setting section not very clear, Kindly make this section clearer. Line 109, How did you ensure participants targeted online through various social media platforms met all your inclusion criteria (i.e infection history, actual current location, and country (your questionnaire doesn't capture information on the town, region, or city), and met the age requirements)

Line 121, Kindly clarify the sampling methods used. what is your rationale behind the selection of particular participants?

Since you collected data both physically and through Social media, it will be interesting to present analyses for respondents by channel of response (physical and social Media), as this might have some correlation with outcomes.

Line 237, Figure 3 showing analysis of vaccine hesitancy is missing

Line 332 - 333, You mention vaccine hesitance was for a variety of reasons and go ahead to site just two reasons. Good elaboration on the various objectives in your discussion section. I recommend you do the same for Vaccine hesitancy

Reviewer #2: 1. The topic is interesting but the author has raise a lot of concepts that he could not full address. To talk about Pre- and Post-COVID-19 Diagnosis, Treatment, Symptoms, Face mask

Ownership/Utilisation, COVID-19 Preventive Measures, and COVID-19 Vaccine Hesitancy is too much that can be full addressed by your finding. consider reducing the scope of your topic.

2. Your background doesn't showcase the problem you are addressing and the figures presented are outdated.

3. Your interpretation of OR are not correct. When your confidence interval crosses 1 its means there is no relationship.

4. All concepts handle under your study should be clearly discussed and probably under different paragraphs

6. PLOS authors have the option to publish the peer review history of their article (what does this mean?). If published, this will include your full peer review and any attached files.

Reviewer #1: No

Reviewer #2: **Yes: **Charles Njumkeng

---

## [Author Response · Author response to Decision Letter 0]

30 Sep 2022

We wish to immensely thank the Editor and Reviewers for the comments and pointers.

We have objectively responded to the comments and recommendations made in the 'Response to Reviewers'.

---

## [Decision Letter · Decision Letter 1]

28 Oct 2022

PONE-D-22-15424R1Pre- and Post-COVID-19 Diagnosis/Treatment Symptoms, Face mask Ownership/Utilisation, COVID-19 Preventive Measures, and COVID-19 Vaccine Hesitancy Amongst Patients Recovering from the COVID-19 infection in Cameroon: A Cross-Sectional StudyPLOS ONE

Dear Dr. Cho,

Thank you for submitting your manuscript to PLOS ONE. After careful consideration, we feel that it has merit but does not fully meet PLOS ONE’s publication criteria as it currently stands. Therefore, we invite you to submit a revised version of the manuscript that addresses the points raised during the review process. Please submit your revised manuscript by Dec 12 2022 11:59PM. If you will need more time than this to complete your revisions, please reply to this message or contact the journal office at plosone@plos.org. Please include the following items when submitting your revised manuscript:A rebuttal letter that responds to each point raised by the academic editor and reviewer(s). You should upload this letter as a separate file labeled 'Response to Reviewers'.A marked-up copy of your manuscript that highlights changes made to the original version. You should upload this as a separate file labeled 'Revised Manuscript with Track Changes'.An unmarked version of your revised paper without tracked changes. You should upload this as a separate file labeled 'Manuscript'.If applicable, we recommend that you deposit your laboratory protocols in protocols.io to enhance the reproducibility of your results. Protocols.io assigns your protocol its own identifier (DOI) so that it can be cited independently in the future. For instructions see: https://journals.plos.org/plosone/s/submission-guidelines#loc-laboratory-protocols. Additionally, PLOS ONE offers an option for publishing peer-reviewed Lab Protocol articles, which describe protocols hosted on protocols.io. Read more information on sharing protocols at https://plos.org/protocols?utm_medium=editorial-email&utm_source=authorletters&utm_campaign=protocols.

We look forward to receiving your revised manuscript.

Kind regards,

Muhammad Shahzad Aslam, Ph.D.,M.Phil., Pharm-D

Academic Editor

PLOS ONE

Journal Requirements:

Reviewers' comments:

Reviewer's Responses to Questions

**Comments to the Author**

1. If the authors have adequately addressed your comments raised in a previous round of review and you feel that this manuscript is now acceptable for publication, you may indicate that here to bypass the “Comments to the Author” section, enter your conflict of interest statement in the “Confidential to Editor” section, and submit your "Accept" recommendation.

Reviewer #1: All comments have been addressed

Reviewer #2: (No Response)

2. Is the manuscript technically sound, and do the data support the conclusions?

Reviewer #1: Yes

Reviewer #2: Partly

3. Has the statistical analysis been performed appropriately and rigorously? 

Reviewer #1: Yes

Reviewer #2: Yes

4. Have the authors made all data underlying the findings in their manuscript fully available?

Reviewer #1: Yes

Reviewer #2: Yes

5. Is the manuscript presented in an intelligible fashion and written in standard English?

Reviewer #1: Yes

Reviewer #2: No

6. Review Comments to the Author

Reviewer #1: This article is now fit for publishing as all the concerns raised have been addressed by the authors have.

Reviewer #2: 1. The author did not response to my suggestion of reshaping the topic to make it more attractive and better support the finding. It will be good for him to provide clear justification why the topic should be maintained as it is.

2. Author did not rephrase the sentence “symptoms severity range from asymptomatic to moderate” in the abstract. Asymptomatic means no symptom. Kindly indicate the symptoms to ease understanding of which was severe and which was mild

3. Line 103: The author has not explained how he ensured that those surveyed from April to July 4 were not surveyed in August since the study was conducted in two sections. What is a repeated survey?

4. Line 161, Author talk about wearing of MF correctly but he has not explained how he assessed this for those who participate in the survey by responding online questionnaire

5. Line 206 and 207: Please check this out it is not possible for the number of persons (665)enrolled to be lower than the number households(2,126)

6. Line 261-264 the use of more likely indicate association, which is not consistent with your p-values. The still need to be addressed. Same with line 270

7. PLOS authors have the option to publish the peer review history of their article (what does this mean?). If published, this will include your full peer review and any attached files.

Reviewer #1: **Yes: **Nsai Frankline Sanyuy

Reviewer #2: No

---

## [Author Response · Author response to Decision Letter 1]

30 Oct 2022

Face mask Ownership/Utilisation and COVID-19 Vaccine Hesitancy Amongst Patients Recovering from COVID-19 in Cameroon: A Cross-sectional Study

---

## [Decision Letter · Decision Letter 2]

28 Nov 2022

PONE-D-22-15424R2Face mask Ownership/Utilisation and COVID-19 Vaccine Hesitancy Amongst Patients Recovering from COVID-19 in Cameroon: A Cross-sectional StudyPLOS ONE

Dear,

Thank you for submitting your manuscript to PLOS ONE. After careful consideration, we feel that it has merit but does not fully meet PLOS ONE’s publication criteria as it currently stands. Therefore, we invite you to submit a revised version of the manuscript that addresses the points raised during the review process. The strength and limitation of study must present after discussion. So, please move its current place. Please explain more about limitation of the study to explain the potential bias in the study.

There is no study implication given. Please give in detail at the end.

Please provide the response rate of the study.

Please provide study information sheet as supplementary file.

Please explain in details about the query of the reviewer "Line 206 and 207: Please check this out it is not possible for the number of persons (665)

enrolled to be lower than the number households (2,126)."

Please explain in details on comments given by reviewer. The peer reviewer are not satisficed with comments and does not explain well. Please include your questionnaire inside the manuscript and explain all comments in manuscript in detail.

Please provide strobe statement and strobe checklist as supplementary file.

Please provide raw data as well as supplementary file.

We look forward to receiving your revised manuscript.

Kind regards,

Muhammad Shahzad Aslam, Ph.D.,M.Phil., Pharm-D

Academic Editor

PLOS ONE

---

## [Author Response · Author response to Decision Letter 2]

1 Dec 2022

Line 206 and 207: Please check this out it is not possible for the number of persons (665) enrolled to be lower than the number households (2,126).

We interviewed one person per household; only one who had suffered from COVID-19 and got treated. In the questionnaire, we enquired the number of persons living in the household (Question 7 on the questionnaire).

Thus, the statement is, 665 persons/household included in the study, 2,126 de facto household residents were counted. This implies that 2,126 household residents were counted in the 665 households. Thus, 1,461 (2,126 - 665) persons were exposed.

Due to adjustments, the lines have changed to lines 203 – 204.

The above issue and others have been raised and we have objectively provided answers.

---

## [Decision Letter · Decision Letter 3]

26 Dec 2022

Face mask Ownership/Utilisation and COVID-19 Vaccine Hesitancy Amongst Patients Recovering from COVID-19 in Cameroon: A Cross-sectional Study

PONE-D-22-15424R3

Dear Dr. Cho,

We’re pleased to inform you that your manuscript has been judged scientifically suitable for publication and will be formally accepted for publication once it meets all outstanding technical requirements.

Kind regards,

Muhammad Shahzad Aslam, Ph.D.,M.Phil., Pharm-D

Academic Editor

PLOS ONE

Additional Editor Comments (optional):

Reviewers' comments:

Reviewer's Responses to Questions

**Comments to the Author**

1. If the authors have adequately addressed your comments raised in a previous round of review and you feel that this manuscript is now acceptable for publication, you may indicate that here to bypass the “Comments to the Author” section, enter your conflict of interest statement in the “Confidential to Editor” section, and submit your "Accept" recommendation.

Reviewer #1: All comments have been addressed

2. Is the manuscript technically sound, and do the data support the conclusions?

Reviewer #1: Yes

3. Has the statistical analysis been performed appropriately and rigorously? 

Reviewer #1: Yes

4. Have the authors made all data underlying the findings in their manuscript fully available?

Reviewer #1: (No Response)

5. Is the manuscript presented in an intelligible fashion and written in standard English?

Reviewer #1: Yes

6. Review Comments to the Author

Reviewer #1: All comments have been addressed by the authors as requested. There are no further comments to present to the authors

7. PLOS authors have the option to publish the peer review history of their article (what does this mean?). If published, this will include your full peer review and any attached files.

Reviewer #1: **Yes: **Nsai Frankline Sanyuy

---

## [Editor Report · Acceptance letter]

6 Jan 2023

PONE-D-22-15424R3 

Face mask Ownership/Utilisation and COVID-19 Vaccine Hesitancy Amongst Patients Recovering from COVID-19 in Cameroon: A Cross-sectional Study 

Dear Dr. Cho:

I'm pleased to inform you that your manuscript has been deemed suitable for publication in PLOS ONE. Congratulations! Your manuscript is now with our production department. 

Kind regards, 

on behalf of

Dr. Muhammad Shahzad Aslam 

Academic Editor

PLOS ONE